# Chunking: Forgetting Matters in Continual Learning even without Changing Tasks

## Abstract

Work on continual learning (CL) has largely focused on the problems arising from the dynamically-changing data distribution. However, CL can be decomposed into two sub-problems: (a) shifts in the data distribution, and (b) dealing with the fact that the data is split into chunks and so only a part of the data is available to be trained on at any point in time. In this work, we look at the latter sub-problem—the *chunking* of data—and note that previous analysis of chunking in the CL literature is sparse. We show that chunking is an important part of CL, accounting for around half of the performance drop from offline learning in our experiments. Furthermore, our results reveal that current CL algorithms do not address the chunking sub-problem, only performing as well as plain SGD training when there is no shift in the data distribution. We analyse why performance drops when learning occurs on chunks of data, and find that forgetting, which is often seen to be a problem due to distribution shift, still arises and is a significant problem. Motivated by an analysis of the linear case, we show that per-chunk weight averaging improves performance in the chunking setting and that this performance transfers to the full CL setting, where there is distribution shift. Hence, we argue that work on chunking can help advance CL in general.

## 1 Introduction

How should we update a neural network efficiently when we observe new data? This issue remains an open problem, and is one that the field of *Continual learning* (CL) addresses. Many methods (Delange et al., 2021; Parisi et al., 2019; Wang et al., 2023) and setups (Hsu et al., 2018; Antoniou et al., 2020; van de Ven & Tolias, 2019) have been proposed in recent years. Specifically, CL studies settings where a learner sees a stream of chunks of data and where the data distribution for each chunk changes over time. This type of change in the data distribution is known as *task shift* (Caccia et al., 2020). Performing continual learning is thwarted by a persistent problem; information learnt from previously seen data is forgotten when updating on a new chunk of data (Kirkpatrick et al., 2017). Tackling this problem is then one of the main focuses of CL research.

CL can be decomposed into two sub-problems: (a) learning with a changing data distribution, and (b) only having access to a single chunk of data for learning at any point in time, unable to ever re-access previous chunks. We call this latter sub-problem the *chunking problem* and analyse it in this work. We show it is responsible for a significant part of the performance difference between CL and offline learning—learning with full access to all the data. Also, our experiments demonstrate that current methods for CL do not counter this sub-problem at all, performing comparably to plain SGD training in the task-shift-free *chunking setting*. Therefore, we suggest that chunking has been underlooked as a problem in CL and in this work we set out to address this imbalance by looking at it in more detail.

Our analysis of the chunking setting establishes a number of findings. First, we show that the size of each chunk has a significant impact on performance: learning on small chunks leads to much worse performance. Second, our experiments demonstrate that forgetting is the main reason for the performance drop in the chunking setting compared to offline learning. This casts doubt on the common sentiment that forgetting is caused mainly by task shift (Lee et al., 2021; Ramasesh et al., 2020). Third, motivated by an analysis of the linear case in the chunking setting, we look at per-chunk weight averaging which improves performance in the chunking setting and reduces the

amount of forgetting. We also show that this performance benefit transfers to the full CL setting—where there is also task shift—establishing that work on the chunking sub-problem has the potential to impact CL in general.

The main contributions of this work are:

- Reviving awareness that online training in neural networks is itself an issue, irrespective of task shift. In the context of CL, we formulate that as *the chunking problem*, and demonstrate it is the reason for a large part of the performance drop between offline learning and CL.

- Analysis of chunking, where we show among other things that forgetting is a key problem and that current CL methods do not improve performance in the chunking setting.

- Proposal of a simple method, per-chunk weight averaging, which improves performance under chunking significantly. Furthermore, this performance transfers to the full CL setting, demonstrating how work on chunking can help improve CL in general.

## 2 PRELIMINARIES AND RELATED WORK

Continual Learning (CL) is a well-studied problem, with many different settings and methods being proposed (van de Ven & Tolias, 2019; Wu et al., 2022; Mirzadeh et al., 2020; Delange et al., 2021). We focus on classification problems. In this context, standard CL (sometimes called offline CL (Prabhu et al., 2020)), consists of a learner seeing a sequence of tasks. Each *task* consists of a single chunk of data with all the training data from a subset of classes in the dataset (van de Ven & Tolias, 2019). A learner only views each task once and can only revisit data from previous tasks which it has stored in a limited memory buffer. For example, for CIFAR-10 (Krizhevsky, 2009) a learner might first see all the data for the airplane and ship classes, then see the data from the dog and cat classes and so on, seeing all data from two classes at a time until the learner has seen all the classes. In addition to standard CL, there is another common CL setting called online CL (Mai et al., 2021; Lee & Storkey, 2023). As shown in Figure 1, in online CL instead of there being a one-to-one map between tasks and chunks, the data for a task is split into multiple smaller chunks, the size of mini-batches, and a learner only sees each chunk once and so cannot revisit previous chunks even though they are of the same task.

In this work we look at the chunking sub-problem of CL. This problem is closely related to online learning, without task shift (Hoi et al., 2021; Bottou & Le Cun, 2003). In both cases the data is observed in the form of a stationary data stream. However, in chunking the data is batched into chunks to match modern neural network learning processes. Straight online learning can be seen as a special case when each chunk consists of one data instance. Furthermore, we investigate the neural network case in contrast to much work in online learning which focuses on the linear case (Hoi et al., 2021). There is recent work on online learning of neural networks, for example Ash & Adams (2020); Caccia et al. (2022); and Sahoo et al. (2017). But, they do not link or compare their work to CL and often the settings and assumptions are quite different from CL. This is unlike our work which focuses on providing insight into CL, which to the best of our knowledge has not been looked at in detail before.

As part of our analysis of the chunking setting we observe that preventing forgetting is the main challenge. The problem of forgetting in online learning of neural networks (without task shift) has a long history. The term *catastropic forgetting* originates in work on Hopfield associative memories (Hopfield, 1982), where online addition of new data eventually results in the erasure of all stored memories, and minimising forgetting to increase storage was a goal (Storkey, 1997). The general problem of forgetting during learning was subsequently characterised by Grossberg (1988) as the *stability–plasticity dilemma*. In the following decade, the issue of forgetting in online learning of feedforward networks (de Angulo & Torras, 1995; Polikar et al., 2001) also received some attention. Despite not being a solved problem, it became less of a focus as non-neural approaches for machine learning came to the fore in the mid 1990s. With a resurgence of interest in neural networks, online learning was reconsidered in the form of CL (Delange et al., 2021; Ramasesh et al., 2020; Wang et al., 2023), but with a focus on more realistic settings that also involve task shift (Ramasesh et al., 2020; Lee et al., 2021). Because of this greater complexity, the component of forgetting due to incremental online learning or chunking has been comparatively understudied in the CL literature.

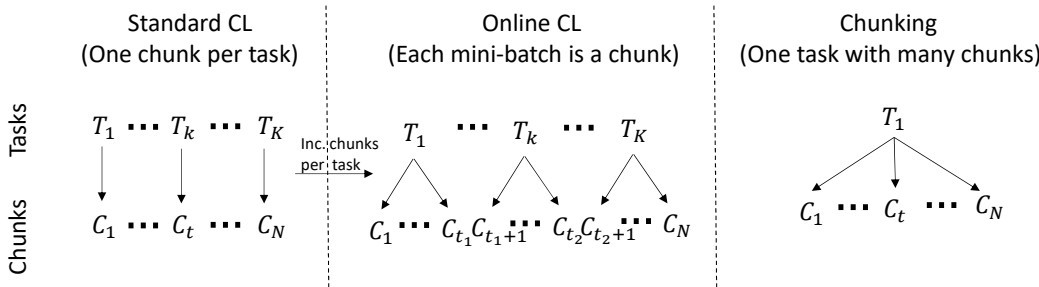

Figure 1: Diagram showing the standard CL, Online CL and chunking settings, where $T_k$ denotes a task, $C_t$ denotes a chunk and the arrows indicate which chunk belongs to which task. The figure shows that the chunking setting is a reduced CL setting where the task does not change.

Yet decomposing a problem can aid its solution, and indeed, we show that chunking is responsible for a large part of the forgetting that happens in CL.

To improve performance in the chunking setting we look at using per-chunk weight averaging. There have been many weight averaging approaches proposed for offline learning (Izmailov et al., 2018; Tarvainen & Valpola, 2017). Instead, in this work we look at applying a per-chunk weight averaging approach to the online chunking setting, motivated by our analysis of the linear case.

## 3 THE CHUNKING SETTING

In the *chunking setting*, a learner sees a sequence of chunks $C_1, C_2, \ldots, C_N$, and trains on one chunk of data at a time: chunks are not revisited. Each chunk of data consists of instance pairs $(x, y)$, with $x \in X$ (e.g. images) and labels $y \in Y$. The data in all chunks are drawn from the same distribution, so there is no distribution shift. Furthermore, in this paper, to control for class imbalance effects we consider a *balanced* chunking setting (henceforth assumed); we constrain each chunk to have as close to the same number of instances for each class in $Y$ as possible. In this way we ensure the results of our experiments are solely due to the effects of limited data availability through chunking and not due to class imbalance. We record results for the case where the chunks are class imbalanced in Appendix B and observe that for our experimental setup class imbalance does not have any significant effect.

In practice, to perform experiments in the chunking setting, consider a class-balanced training dataset of size $M$ and a target chunk size $S$. First, we randomly reorder the training data for each class, and then arrange all the data into a class-ordered list. Data is then sequentially allocated into $\lfloor M/S \rfloor$ chunks by assigning each element of the list in turn to a chunk, in a cyclical fashion. So, the first data item goes into chunk 1, second into chunk 2 etc., up to an item into chunk $\lfloor M/S \rfloor$, then the next into chunk 1 again and so on. Then we randomly permute the data within each chunk and randomly reorder the chunks themselves. To ensure chunks are fully balanced, in the experiments in this paper we choose chunk sizes so that all chunks are of equal size and contain the same number of data instances for each class. Finally, we reserve a fixed-sized portion of data from each class to form a test set which is used to evaluate the accuracy of a method.

The only difference between the chunking setting and the full CL setting is the lack of task shift. We do not change tasks in the chunking setting and instead the stream consists of chunks from a single task which contains all the training data from all the classes, as shown in Figure 1. Therefore, the chunking setting provides a simple way to analyse and understand the problems caused by chunking itself. Also, performance in the chunking setting gives an upper bound to CL performance, and so without solving this setting CL will never be able to improve beyond current chunking performance.

## 4 ANALYSIS OF THE CHUNKING SETTING

To see how much chunking impacts the performance in CL, we look at its relative contribution to the performance drop of CL from offline learning, showing it plays a significant part. This was

Table 1: Accuracy of DER++ when using a ResNet18 in the offline, chunking and standard CL class-incremental settings, along with the percentage drop in accuracy from offline learning to CL due to chunking (Chunking Prop.). We split each dataset into 10 tasks following the experimental setup of Buzzega et al. (2020) and Boschini et al. (2022) and also use a memory size selected in those works.

| Dataset | Memory Size | Offline | Chunking | CL | Chunking Prop. |
|---|---|---|---|---|---|
| CIFAR-100 | 2000 | $73.72_{\pm0.115}$ | $63.35_{\pm0.348}$ | $53.00_{\pm0.327}$ | $50.05\%$ |
| Tiny ImageNet | 5120 | $60.63_{\pm0.366}$ | $50.54_{\pm0.118}$ | $39.02_{\pm0.97}$ | $46.69\%$ |

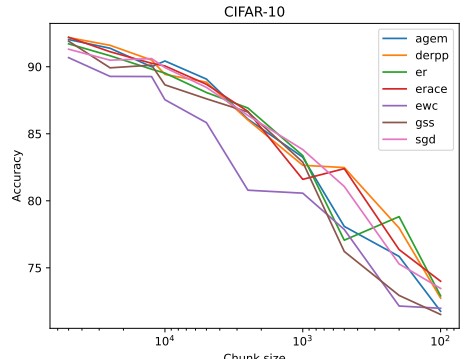 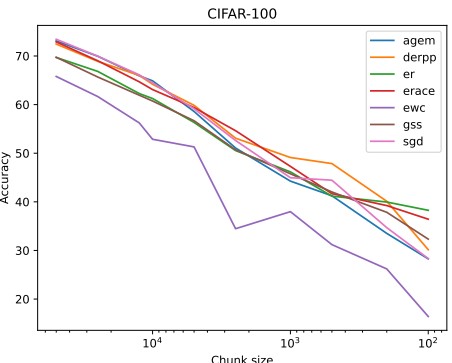

Figure 2: End-of-training accuracy against chunk size on CIFAR-10 and CIFAR-100, where each data point on a curve presents the end-of-training accuracy of a method from a full run with chunks of the size given on the horizontal axis.

achieved by performing an experiment where we compare the performance of the state-of-the-art CL method DER++ (Buzzega et al., 2020) for both standard CL and the chunking setting to offline SGD training. Where for standard CL we look at class-incremental learning which means that at test time, the learner has to classify across all the classes (van de Ven & Tolias, 2019), like the chunking setting. For the experiments, we use a ResNet18 backbone and a 10 task/chunk split of CIFAR-100 (Krizhevsky, 2009) and Tiny ImageNet (Stanford, 2015)—the rest of the experimental details are given in Appendix A. The results are presented in Table 1 and show that the performance drop between offline learning and chunking is $50.05\%$ and $46.69\%$ of the full performance drop from offline learning to CL for CIFAR-100 and Tiny ImageNet, respectively. This indicates that a significant part of the performance drop of CL from offline learning is due to chunking and not due to the task changing. Also, in the real world it is often the case that the hard task shifts commonly used in continual learning do not happen (Bang et al., 2021; 2022) and instead there are smoother changes between tasks which should reduce the effect of task shift and increase the importance of dealing with chunking.

## 4.1 Performance in the Chunking setting

Our results on the chunking setting show that CL methods perform no better than plain SGD and perform worse as the size of the chunks decreases. For instance, Figures 2 and 3 present the performance of state-of-the-art CL methods for different chunk sizes and a memory buffer size of 500 examples on CIFAR-10, CIFAR-100 and Tiny Imagenet, which are commonly used in the CL literature (Delange et al., 2021). We train on each chunk for 50 epochs for CIFAR-10 and CIFAR-100 and 100 epochs for Tiny ImageNet which we found to be give the best or comparable to the best performance (as shown in Appendix C). The full experimental details of this experiment are described in Appendix A. The results show that there is a large performance drop as the chunk size decreases. For example, on CIFAR-100 for offline learning when all the data is in one chunk, corresponding to a chunk size of 50000, CL methods get a test accuracy of around 73% but when each chunk consists of 1000 examples they get around $45\%$. Also, Figures 2 and 3 show that all the CL methods perform

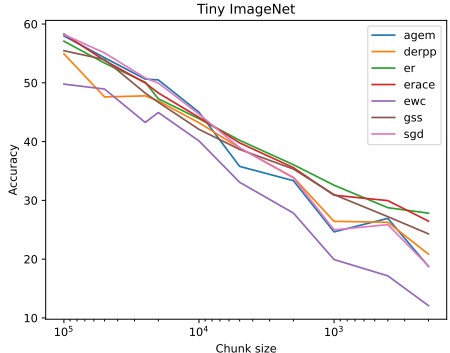 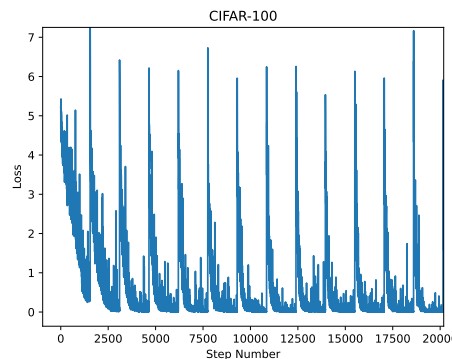

Figure 3: End-of-training accuracy against chunk size for Tiny ImageNet, where each data point on a curve presents the end-of-training accuracy of a method from a full run on Tiny ImageNet with chunks of the size given on the horizontal axis.

Figure 4: The training loss curve for plain SGD on CIFAR-100 when training on 50 chunks, where we plot the training loss for the first 2000 update steps corresponding to learning on the first 13 chunks.

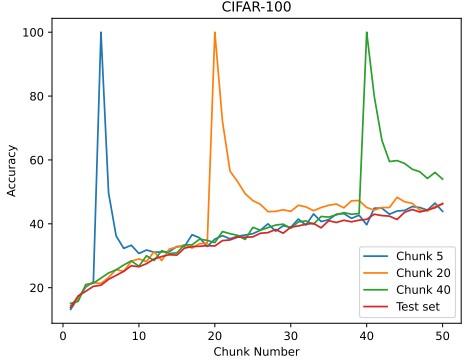 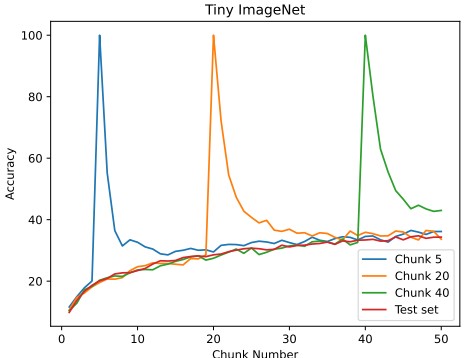

Figure 5: Accuracy at the end of learning on each chunk for the training set of the $5^{th}$, $20^{th}$ and $40^{th}$ chunks and the test set, for CIFAR-100 and Tiny ImageNet. We split the datasets into 50 chunks, corresponding to a chunk size of 1000 and 2000 for CIFAR-100 and Tiny ImageNet, respectively.

roughly the same as plain SGD. Hence, our results indicate current CL methods do not tackle the chunking problem at all and instead have focused on reducing the performance drop due to task shift, as they perform much better than SGD on settings with task shift (Wang et al., 2023). One point to note on this is that the replay methods ER (Chaudhry et al., 2020) and ER-ACE (Caccia et al., 2021) perform better than SGD for very small chunk sizes. This is due to them storing 500 examples in memory and so have an effective chunk size of 500 more data points than SGD, which impacts performance for chunk sizes below 500.

An important question to ask is why does chunking reduce the performance from the offline setting. There are three general possibilities: not integrating all the information a chunk has into the model (underfitting), fully integrating each chunk's information but at the cost of forgetting previous information (forgetting) or a mixture of both. To explore which possibility is true we look at training using 50 chunks and present the training loss curve in Figure 4 of learning on the first 13 chunks for CIFAR-100 and in Figure 5 the test accuracy and accuracy on the training data for the $5^{th}$, $20^{th}$ and $40^{th}$ chunks evaluated at the end of each chunk, for CIFAR-100 and Tiny ImageNet (see Appendix D for CIFAR-10). The training loss curve in Figure 4 shows that we fit each chunk well as the loss plateaus for each chunk and at a relatively low value. Furthermore, the accuracy curves for each chunks training data in Figure 5 establishes that after training on the chunk the model fits it perfectly, achieving an accuracy of $100\%$. Hence, we know that we fit each chunk well, removing the possibility of underfitting. Figure 5 also shows that after learning on the chunk the accuracy on that chunks data quickly drops and falls back to the level of the test set performance, showing that the learner is forgetting a lot of the chunk's information. However, not all of a chunk's information

is forgotten as the test accuracy improves as the learner sees more chunks. Therefore, our results establish that the performance drop in the chunking setting is due to forgetting. This demonstrates that forgetting is not only due to task shift, like suggested in previous work (Lee et al., 2021; Ramasesh et al., 2020), but that it is also due to seeing the data in chunks.

We have shown that forgetting is the reason for the reduced performance in the chunking setting, when compared to offline learning. However, not all the information provided by a chunk is forgotten and if a learner could repeatedly resample chunks it would approach offline performance. This fact is standard knowledge for online learning (Bottou & Le Cun, 2003) and has recently been shown to be true for CL with task-shift (Lesort et al., 2023). However, unlike standard online learning, in the chunking setting and CL in general it is not possible to resample chunks and so in these settings we need to be able to fully learn a chunk of data without needing to repeatedly revisit it in the future. This implies that improving chunking performance and reducing forgetting is closely related to improving the efficiency of learning. Hence, we hope that work on improving chunking performance will also improve the general efficiency of learning algorithms and vice versa.

## 4.2 ANALYSIS OF THE LINEAR CASE

To analyse the chunking problem further we turn to the linear regression case, where we can leverage closed form solutions. In this case, the naive solution is to perform least squares on each arriving chunk. However, as the least squares problem is convex and so does not depend on the initialised weights, it will fully forget all the past chunks, only using the last chunk to create the predictor. This means that the standard least squares solution to linear regression fails in the chunking setting. Instead a better solution is to use Bayesian linear regression (Minka, 2000). This is because Bayesian linear regression given any particular chunking of the data will return the same predictor and so fully solves the chunking setting. Therefore, it is instructive to see how Bayesian linear regression prevents forgetting. To achieve this we present below the update equations for Bayesian linear regression. The prior on the weights is $\boldsymbol{\theta} \sim \mathcal{N}(\mathbf{0}, \mathbf{V}_0)$ and the posterior after seeing all the chunks up to and including the $(k-1)$th is $\boldsymbol{\theta}|C_{1:k-1} \sim \mathcal{N}(\mathbf{m}_{k-1}, \mathbf{V}_{k-1})$. Additionally, for a chunk $C_t$ we define $\mathbf{X}_t$ as its row-wise matrix of data instances and $\mathbf{y}_t$ as its vector of targets. The likelihood is defined by assuming $y|\mathbf{x}, \boldsymbol{\theta} \sim \mathcal{N}(\boldsymbol{\theta}^T\mathbf{x}, \sigma^2)$. Then, the Bayesian posterior on the $k$th chunk is

$$\boldsymbol{\theta}|C_{1:k} \sim \mathcal{N}(\mathbf{m}_k, \mathbf{V}_k), \tag{1}$$

$$\mathbf{m}_k = \mathbf{V}_k \mathbf{V}_{k-1}^{-1} \mathbf{m}_{k-1} + \frac{1}{\sigma^2} \mathbf{V}_k \mathbf{X}_k^T \mathbf{y}_k, \tag{2}$$

$$\mathbf{V}_k^{-1} = \mathbf{V}_k^{-1} + \frac{1}{\sigma^2} \mathbf{X}_k^T \mathbf{X}_k. \tag{3}$$

By recursively expanding the $\mathbf{m}_{k-1}$ and $\mathbf{V}_{k-1}$ terms till we reach the prior we have that

$$\mathbf{m}_k = \frac{1}{\sigma^2} \sum_{t=1}^{k} \mathbf{V}_k \mathbf{X}_t^T \mathbf{y}_t, \tag{4}$$

$$\mathbf{V}_k^{-1} = \mathbf{V}_0^{-1} + \frac{1}{\sigma^2} \sum_{t=1}^{k} \mathbf{X}_t^T \mathbf{X}_t = \mathbf{V}_0^{-1} + \frac{1}{\sigma^2} \mathbf{X}_{1:k}^T \mathbf{X}_{1:k}. \tag{5}$$

The equations above show that Bayesian linear regression prevents forgetting by having its posterior mean $\mathbf{m}_k$ be: (a) a sum of the least squares solutions of each chunk and (b) instead of using the chunks unnormalised empirical covariance $\mathbf{X}_t^T\mathbf{X}_t$ in the least squares solutions it uses the running estimate of the weight precision $\mathbf{V}_k^{-1}$. Computing and storing $\mathbf{V}_k^{-1}$ is infeasibly costly for very large systems (e.g. neural networks), taking up $O(\dim(\boldsymbol{\theta})^2)$ space. Therefore, assuming there is only enough memory to store a set of weights a backoff is to use a sum of the least squares solutions to each chunk. This is achieved by *weight averaging*, where at each chunk we perform least squares on that chunk and add it to a running average, which results in the update equation,

$$\mathbf{m}_k = \frac{k-1}{k} \mathbf{m}_{k-1} + \frac{1}{k\sigma^2} (\mathbf{X}_k^T \mathbf{X}_k)^{-1} \mathbf{X}_k^T \mathbf{y}_k. \tag{6}$$

Again, by recursively expanding $\mathbf{m}_{k-1}$ we have that,

$$\mathbf{m}_k = \frac{1}{k\sigma^2} \sum_{t=1}^{k} (\mathbf{X}_t^T \mathbf{X}_t)^{-1} \mathbf{X}_t^T \mathbf{y}_t. \tag{7}$$

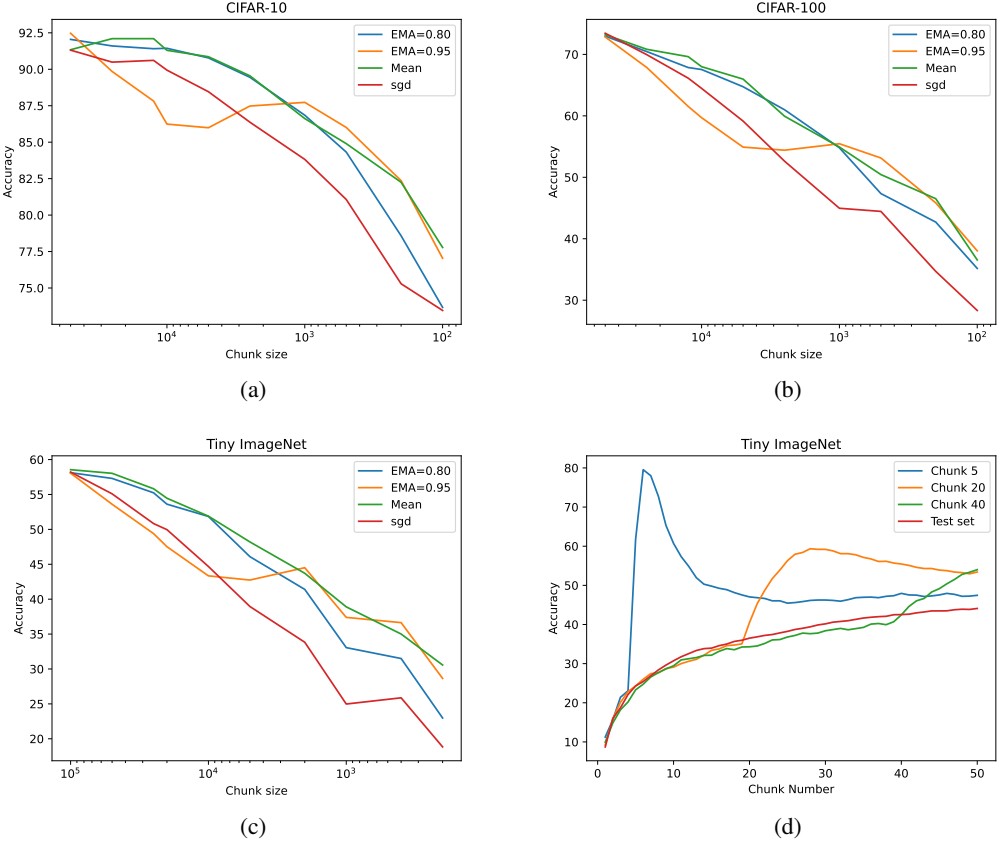

(a)      (b)

(c)      (d)

Figure 6: Plots (a), (b) and (c) show the end-of-training accuracy when leaning with the given chunk size for CIFAR-10, CIFAR-100 and Tiny ImageNet, where sgd is learning without weight averaging and we display EMA results for $\alpha=0.8$ and $0.95$. Plot (d) shows when using mean weight averaging the accuracy at the end of learning on each chunk for the training set of the $5^{th}$, $20^{th}$ and $40^{th}$ chunks and the test set, for Tiny ImageNet with 50 chunks, corresponding to a chunk size of 2000.

Weight averaging gives similar, mean, weights as Bayesian linear regression where instead of using $\mathbf{V}_k$ it uses the per-chunk estimate $\frac{1}{k}(\mathbf{X}_t^T \mathbf{X}_t)^{-1}$ and we divide by $k$ to correctly scale the estimate. Both $\mathbf{V}_k$ and $\frac{1}{k}(\mathbf{X}_t^T \mathbf{X}_t)^{-1}$ are unnormalised estimates of the precision of the data distribution. Therefore, when each chunk is large enough that they are both accurate estimates, we have that $\frac{1}{k}(\mathbf{X}_t^T \mathbf{X}_t)^{-1} \approx \mathbf{V}_k$ for all $t \in \{1, \dots, k\}$. In this case, weight averaging approximates Bayesian linear regression well and so should not forget that much. This means it greatly improves performance over standard linear regression, which forgets all but the last chunk. Hence, the question arises if this analysis showing weight averaging improving performance, assuming the chunks are large enough, still holds true for neural networks. We look at this in the next section.

## 5    PER-CHUNK WEIGHT AVERAGING

From the analysis of the linear case (Section 4.2), we see that averaging the weights learnt at the end of each chunk is a way to improve performance in the chunking setting for linear models. Motivated by this, we now look at the neural network case, where our results show that weight averaging also improves performance, often by a large margin. More precisely, the simple method we look at, calling it per-chunk weight averaging, consists of training the model as normal but we additionally store an average of the weights learnt at the end of each chunk, which is not used in training but in evaluation is used as the weights of the network. Here we consider the *weights* to be all the parameters of the neural network, including batch normalisation statistics (Ioffe & Szegedy, 2015).

More specifically, we look at using in evaluation the mean or an exponential moving average (EMA) of the weights found after training on each chunk up to some chunk $k$, defined by

$$\boldsymbol{\theta}_k^{MEAN} = \frac{1}{k}\sum_{t=1}^{k}\boldsymbol{\theta}_t \qquad (8)$$

$$\boldsymbol{\theta}_k^{EMA} = \alpha\boldsymbol{\theta}_{k-1}^{EMA} + (1-\alpha)\boldsymbol{\theta}_k, \qquad (9)$$

where $\boldsymbol{\theta}_t$ is the value of the weights after learning on chunk $C_t$ and for EMA, $\alpha \in [0,1]$ controls how much weight is given to old versus newly learnt end-of-chunk weights.

To observe whether per-chunk weight averaging improves performance in the chunking setting, we carry out experiments using it in combination with plain SGD training. The reason we only look at plain SGD training and not a CL method is that, as shown in Figures 2 and 3, no CL method looked at performs any better than SGD in the chunking setting. The experimental setup is the same as the previous experiments and is described in Appendix A. The results of the experiments are presented in plots (a), (b) and (c) of Figure 6 and show that it is clear that for all three datasets—CIFAR-10, CIFAR-100 and Tiny ImageNet—using a per-chunk weight average in evaluation increases accuracy. For instance, for the smallest chunk size looked at for each dataset, using mean weight averaging improves accuracy by $+4.32\%$, $+8.22\%$ and $+11.73\%$ for CIFAR-10, CIFAR-100 and Tiny ImageNet, respectively. Additionally, Figure 6 demonstrates that using the mean is better than or comparable to using EMA for nearly all chunk sizes on each dataset. We only display EMA for two $\alpha$ values in the figure but we looked at many more in Appendix E, and selected the two best values to show in Figure 6. So, our results show that using the mean of the weights learnt after learning on each chunk for prediction is an effective way to improve performance in the chunking setting.

To analyse why per-chunk weight averaging improves performance, we look at how well it preserves the information of past chunks. To do this, like in Figure 5, we measure, for per-chunk mean weight averaging, the test accuracy and the accuracy on the training data of the $5^{th}$, $20^{th}$ and $40^{th}$ chunks at the end of learning on each chunk, when using 50 chunks. The results are shown in plot (d) of Figure 6 for Tiny ImageNet and for CIFAR-10 and CIFAR-100 in Appendix D. By comparing these results to the ones when using the final weights for evaluation, shown in Figure 5, we see that when using per-chunk mean weight averaging more information is preserved from previous chunks. This is because using it gives higher accuracy on the training data from previous chunks than the test set long after that chunk was trained on. While, when using the final weights for evaluation this is not the case, as after learning on a chunk the accuracy on the training data of that chunk drops quickly down to around the test set accuracy. Therefore, part of the reason per-chunk weight averaging performs well is that it forgets less than plain SGD in the chunking setting.

## 5.1 APPLICATION TO CONTINUAL LEARNING

While per-chunk weight averaging improves performance in the chunking setting, it is also important to see how this translates to the full CL setting, so that we can see how work on the chunking setting can impact CL in general. To do this we perform experiments using mean weight averaging in class and task incremental learning (van de Ven & Tolias, 2019), the two main CL scenarios, using four standard well-performing methods: DER++ (Buzzega et al., 2020), experience replay (ER) (Chaudhry et al., 2020), AGEM (Chaudhry et al., 2019) and GSS (Aljundi et al., 2019). As in common with the rest of this work and many works on continual learning (Delange et al., 2021; Buzzega et al., 2020), we use CIFAR-10, CIFAR-100 and Tiny ImageNet as the datasets for this experiment, splitting CIFAR-10 into 5 tasks each containing the data of 2 classes and splitting CIFAR-100 and Tiny ImageNet into 10 tasks each consisting of the data of 10 classes for CIFAR-100 and 20 for Tiny ImageNet. The difference between class and task incremental learning is that at test time for task-incremental learning each method only predicts which class a data instance is between the classes of that data instance's task, while for class-incremental learning the method has to classify between all classes seen. Additionally, we look at both standard and online CL (as defined in Section 2) and set the memory size to be 100 examples for all methods. For standard CL methods can repeatedly iterate over the data of a task, in our experiments for each task we use 50 epochs for CIFAR-10 and CIFAR-100 and 100 epochs for Tiny ImageNet, like previous work (Buzzega et al., 2020).

The results of the experiments on per-chunk mean weight averaging in CL are presented in Table 2 and demonstrate that in general it improves performance. For example, in the standard CL setting

Table 2: Accuracy of CL methods in online and standard CL settings when using per-chunk weight averaging (WA-) or not, averaged over 3 runs and where we report the standard error over the runs.

| Setting | Method | CIFAR-10 | | CIFAR-100 | | Tiny ImageNet | |
|---|---|---|---|---|---|---|---|
| | | Class-IL | Task-IL | Class-IL | Task-IL | Class-IL | Task-IL |
| Online | DER++ | $34.76_{\pm2.20}$ | $78.56_{\pm1.10}$ | $6.73_{\pm0.26}$ | $41.21_{\pm1.34}$ | $5.48_{\pm0.21}$ | $30.95_{\pm0.11}$ |
| | WA-DER++ | $33.46_{\pm0.72}$ | $81.97_{\pm0.25}$ | $12.34_{\pm0.19}$ | $52.34_{\pm0.43}$ | $8.53_{\pm0.05}$ | $39.32_{\pm0.55}$ |
| | $\Delta$Acc | $-1.30$ | $+3.41$ | $+5.61$ | $+11.13$ | $+3.05$ | $+8.37$ |
| | ER | $36.19_{\pm1.19}$ | $81.89_{\pm0.92}$ | $8.45_{\pm0.45}$ | $44.14_{\pm1.31}$ | $5.56_{\pm0.21}$ | $27.23_{\pm0.65}$ |
| | WA-ER | $39.59_{\pm0.60}$ | $84.27_{\pm0.37}$ | $14.01_{\pm0.23}$ | $50.66_{\pm0.77}$ | $7.77_{\pm0.09}$ | $34.26_{\pm0.33}$ |
| | $\Delta$Acc | $+3.40$ | $+2.38$ | $+5.56$ | $+6.52$ | $+2.21$ | $+7.03$ |
| | AGEM | $16.82_{\pm0.61}$ | $70.70_{\pm1.92}$ | $4.70_{\pm0.51}$ | $29.56_{\pm1.93}$ | $3.93_{\pm0.22}$ | $20.53_{\pm1.30}$ |
| | WA-AGEM | $22.59_{\pm1.04}$ | $72.37_{\pm3.03}$ | $10.73_{\pm0.35}$ | $44.68_{\pm0.58}$ | $9.06_{\pm0.47}$ | $34.44_{\pm0.59}$ |
| | $\Delta$Acc | $+5.77$ | $+1.67$ | $+6.03$ | $+20.39$ | $+5.13$ | $+13.91$ |
| | GSS | $27.33_{\pm1.26}$ | $81.28_{\pm1.47}$ | $7.93_{\pm0.16}$ | $49.95_{\pm0.24}$ | $5.59_{\pm0.11}$ | $36.00_{\pm0.49}$ |
| | WA-GSS | $35.03_{\pm0.50}$ | $84.51_{\pm0.41}$ | $8.40_{\pm0.32}$ | $54.68_{\pm0.28}$ | $4.82_{\pm0.06}$ | $42.69_{\pm0.38}$ |
| | $\Delta$Acc | $+7.70$ | $+3.23$ | $+0.47$ | $+4.73$ | $-0.77$ | $+6.69$ |
| Standard | DER++ | $53.18_{\pm0.87}$ | $88.90_{\pm0.30}$ | $16.26_{\pm1.22}$ | $58.92_{\pm0.36}$ | $11.08_{\pm0.38}$ | $34.26_{\pm0.32}$ |
| | WA-DER++ | $49.88_{\pm1.63}$ | $93.25_{\pm0.33}$ | $23.46_{\pm1.48}$ | $72.46_{\pm1.08}$ | $12.39_{\pm0.93}$ | $49.51_{\pm0.69}$ |
| | $\Delta$Acc | $-3.30$ | $+4.35$ | $+7.20$ | $+13.54$ | $+1.31$ | $+15.25$ |
| | ER | $40.01_{\pm0.81}$ | $89.79_{\pm0.75}$ | $11.78_{\pm0.34}$ | $57.80_{\pm1.02}$ | $8.36_{\pm0.16}$ | $31.72_{\pm0.46}$ |
| | WA-ER | $56.49_{\pm0.87}$ | $94.28_{\pm0.17}$ | $24.24_{\pm0.64}$ | $70.07_{\pm0.29}$ | $12.31_{\pm0.19}$ | $46.71_{\pm0.33}$ |
| | $\Delta$Acc | $+16.48$ | $+4.49$ | $+12.46$ | $+12.27$ | $+3.95$ | $+14.99$ |
| | AGEM | $20.19_{\pm0.28}$ | $85.80_{\pm1.18}$ | $9.35_{\pm0.01}$ | $46.99_{\pm0.26}$ | $8.15_{\pm0.05}$ | $24.76_{\pm0.62}$ |
| | WA-AGEM | $38.87_{\pm2.83}$ | $92.06_{\pm0.61}$ | $18.05_{\pm0.68}$ | $65.23_{\pm0.61}$ | $10.42_{\pm0.32}$ | $42.75_{\pm0.25}$ |
| | $\Delta$Acc | $+18.68$ | $+6.26$ | $+8.70$ | $+18.24$ | $+2.27$ | $+17.99$ |
| | GSS | $30.91_{\pm1.02}$ | $86.08_{\pm0.35}$ | $10.74_{\pm0.10}$ | $50.30_{\pm0.28}$ | $8.30_{\pm0.01}$ | $27.55_{\pm1.04}$ |
| | WA-GSS | $51.58_{\pm1.14}$ | $93.75_{\pm0.43}$ | $14.78_{\pm0.57}$ | $69.20_{\pm0.35}$ | $6.13_{\pm0.07}$ | $46.57_{\pm1.16}$ |
| | $\Delta$Acc | $+20.67$ | $+7.67$ | $+4.04$ | $+18.90$ | $-2.17$ | $+19.02$ |

using per-chunk mean weight averaging improves performance on average by $+6.39\%$, $+11.11\%$, $+12.02\%$ and $+11.36\%$ for DER++, ER, AGEM and GSS, respectively. While in the online CL setting it improves performance on average by $+5.05\%$, $+4.52\%$, $+8.82\%$ and $+3.68\%$ for DER++, ER, AGEM and GSS, respectively. However, for DER++ on CIFAR-10 and for GSS on Tiny ImageNet for class-incremental learning per-chunk mean weight averaging does worse than using the final learnt weights. However, as a method will have access to both options when using per-chunk mean weight averaging by validating the performance of each option it is always possible to pick the better one, avoiding any accuracy loss. So, we have shown that per-chunk weight averaging improves performance in the chunking setting and that, in general, this improvement transfers to CL, showing that work on the chunking sub-problem can impact CL research as a whole.

## 6 CONCLUSIONS

In this work we have looked at chunking, bringing awareness to the fact it is an important sub-problem of continual learning (CL), being responsible for a large part of the performance drop between offline and CL performance. We have presented results evidencing that current CL methods do not tackle the chunking problem at all, having comparable performance to plain SGD training in the chunking setting. Additionally, we have demonstrated that the reason for the performance drop in the chunking setting is forgetting, and that the size of each chunk has a significant effect on performance. Motivated by an analysis of the linear case, we also look at using per-chunk weight averaging in the chunking setting, showing that it improves performance. Furthermore, we show that per-chunk weight averaging improves performance of CL methods in the full CL setting, indicating that future work on chunking has the possibility of improving CL as a whole.

REPRODUCIBILITY STATEMENT

To make our experiments reproducible, we provide in the main text descriptions of the experimental setups and provide more specific experimental details in Appendix A. Also, we use only simple changes to the public Mammoth CL library (Buzzega et al., 2020) to run the experiments and provide the code used in the supplementary material.

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

## A    EXPERIMENTAL DETAILS

For all of our results we follow the experimental protocol of Buzzega et al. (2020) and Boschini et al. (2022), and use a modification of the CL library *Mammoth* used in those works to run the experiments. Therefore, for all our experiments we use a ResNet18 (He et al., 2016) as the backbone model. Additionally, we utilize augmentations, applying random crops and horizontal flips to images trained on for all the datasets used. To be able to have a fair comparison in our chunking experiments all methods are trained using SGD and with the same number of epochs: 50 epochs for each chunk for CIFAR-10 and CIFAR-100 and 100 for Tiny ImageNet. We use the same mini-batch size for all experiments, which is 32 examples, and for replay CL methods we use 32 as the replay batch size as well. The hyperparameters of methods were found using a grid search on a validation set and are the same as in Buzzega et al. (2020) and Boschini et al. (2022); however, all the results were newly computed by the authors for this work. For all results on the chunking setting a learning rate of 0.1 was used to ensure a fair comparison between methods and chunk sizes. Lastly, the full list of CL methods evaluated in the chunking setting is: AGEM (Chaudhry et al., 2019), DER++ (Buzzega et al., 2020), ER (Chaudhry et al., 2020), ER-ACE (Caccia et al., 2021), EWC (Kirkpatrick et al., 2017), GSS (Aljundi et al., 2019) and plain SGD training.

## B    ANALYSIS OF STRATIFIED SAMPLING FOR CHUNKS

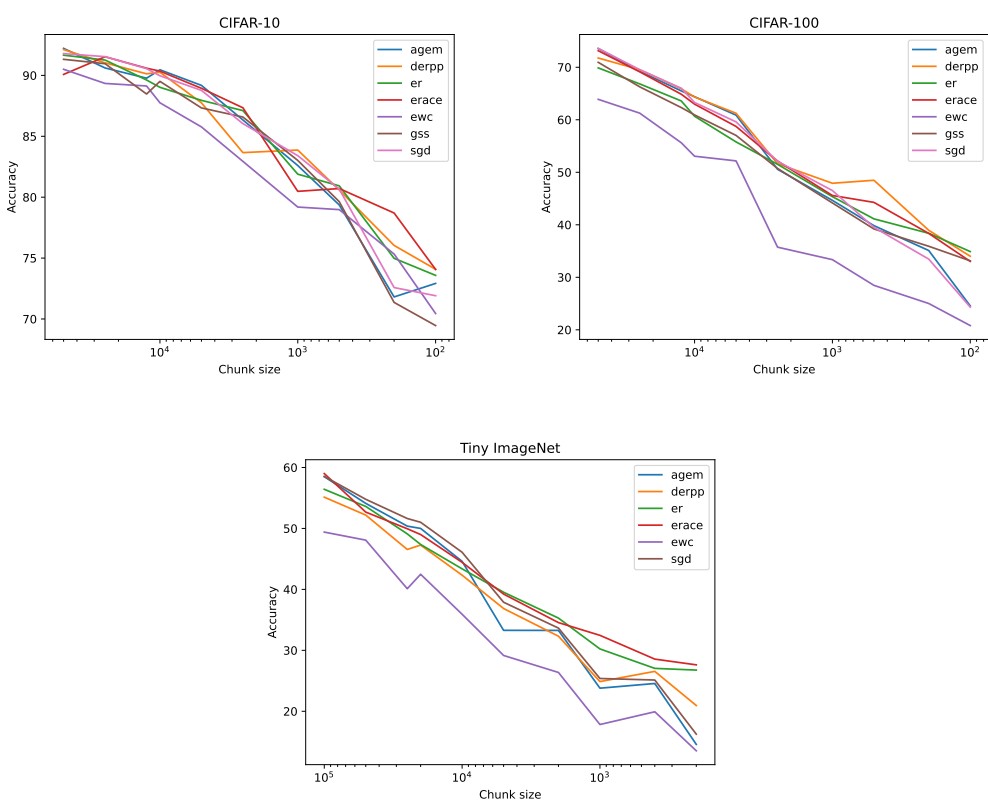

Figure 7: Plots of end-of-training accuracy against chunk size on CIFAR-10, CIFAR-100 and Tiny ImageNet, where the data sampled for each chunk is not constrained to be class balanced. Each data point on a curve presents the end-of-training accuracy of a method from a full run with chunks of the size given on the horizontal axis.

To make sure for small chunk sizes the drop in performance is due to the online availability of data and not class imbalance in a chunk, in the, balanced, chunking setting we stratify sample chunks such that each chunk has an equal amount of data from each class. However, in Figure 7 we look

at what happens if the chunks were sampled randomly without ensuring the classes are balanced in each chunk (i.e., we randomly split the dataset into the given number of chunks). The figure shows the same trend as the when the chunks are classed balanced, shown in Figures 2 and 3, indicating that class imbalance is not a significant problem in the particular instantiation of the chunking setting we look at. However, class imbalance could be a problem for other datasets and is for the linear case therefore we assume the chunks are class balanced in the rest of this work and in our formulation of the chunking setting.

## C   ANALYSIS OF NUMBER OF EPOCHS PER CHUNK

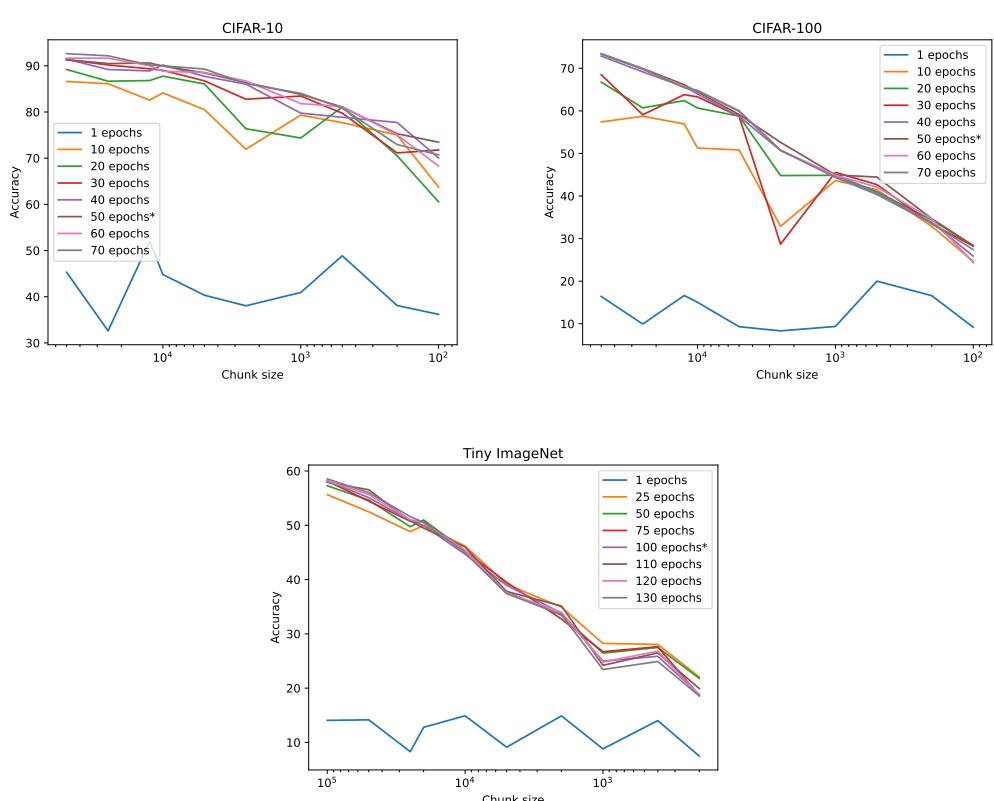

Figure 8: Plots of end-of-training accuracy against chunk size on CIFAR-10, CIFAR-100 and Tiny ImageNet for SGD using different number of epochs per chunk. Each data point on a curve presents the end-of-training accuracy of a method from a full run with chunks of the size given on the horizontal axis. The '*' denotes the number of epochs we use in the rest of the experiments and this figure shows that using our selected values achieves the best, or very similar to the best, accuracy for each chunk size.

An simple fact is that training hyperparameters affect the reason why a method performs badly in the chunking setting. For example, if a method trains for one epoch over each chunk then it probably has bad performance due to underfiting. To remove this dependence we report the behaviour when using the hyperparameters which achieve the best accuracy for each chunk size. In Figure 8 we report the effect the the number of training epochs used for each chunk has on performance, showing our selected number of epochs is the best or comparable to the best for each chunk size. Interestingly, the best performing number-of-epochs is the same for the full CL setting with task shift (Buzzega et al., 2020; Boschini et al., 2022). This suggests that when fitting the training hyperparameters in CL, we are in part implicitly fitting them to minimise the effect of chunking.

# D ADDITIONAL FORGETTING CURVES

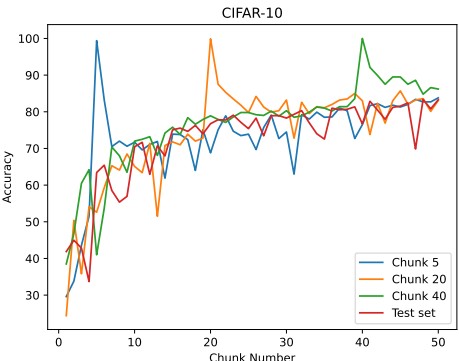

Figure 9: Plot for plain SGD training of accuracy at the end of learning on each chunk for the training set of the $5^{th}$, $20^{th}$ and $40^{th}$ chunks and the test set, when training on CIFAR-10 with 50 chunks, corresponding to a chunk size of 1000.

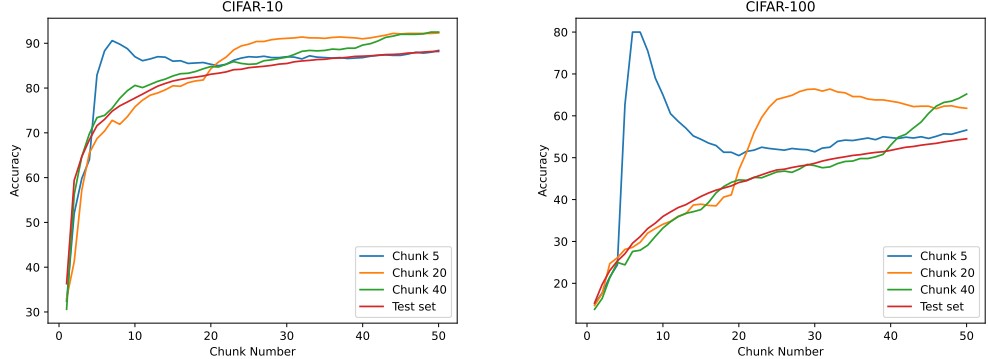

Figure 10: Plot showing when using mean weight averaging the accuracy at the end of learning on each chunk for the training set of the $5^{th}$, $20^{th}$ and $40^{th}$ chunks and the test set, when training on CIFAR-10 and CIFAR-100 with 50 chunks, corresponding to a chunk size of 1000 for both datasets.

# E EXPERIMENT ON DIFFERENT WEIGHTINGS FOR EMA

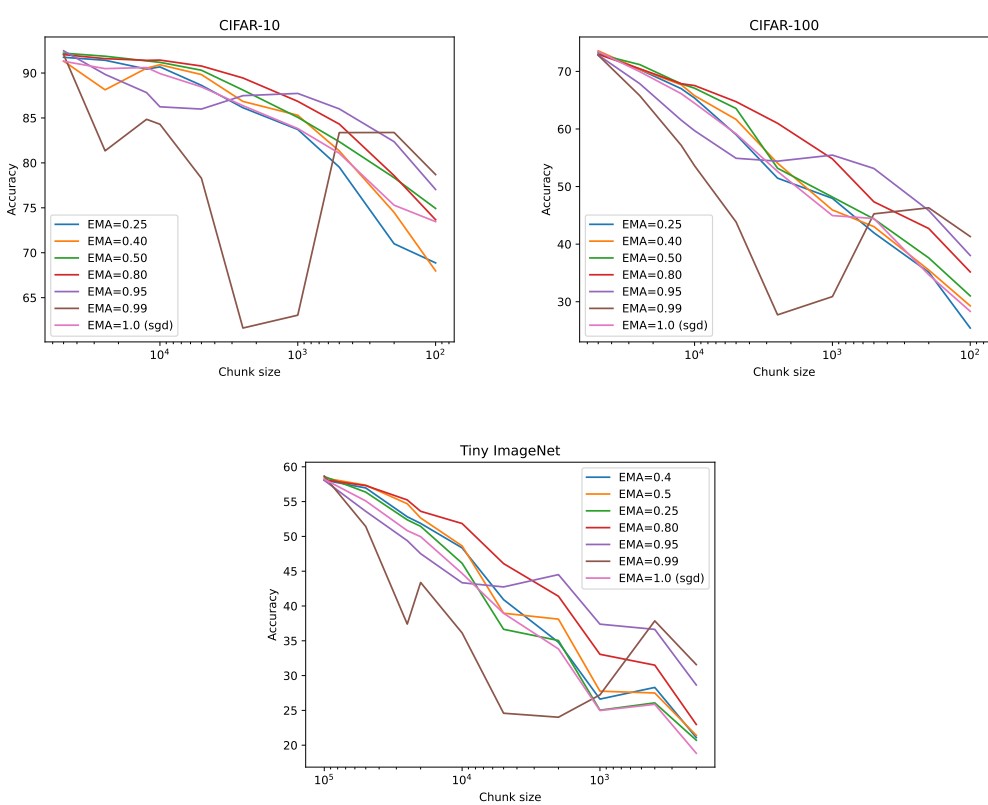

Figure 11: Plots of the end-of-training accuracy when leaning with the given chunk size for different EMA weight values when learning with SGD on CIFAR-10, CIFAR-100 and tiny ImageNet.

