# OpenReview forum: "Chunking: Forgetting Matters in Continual Learning even without Changing Tasks"
_ICLR.cc/2024/Conference — Submitted to ICLR 2024_

### Official Review · Reviewer_BU9q · 2023-10-27

**Soundness:** 2 fair
**Presentation:** 3 good
**Contribution:** 2 fair
**Rating:** 3
**Confidence:** 5

**Summary:**

The paper introduces and explores continual learning setting referred to as "chunking." In this scenario, the learner is exposed to data segments, each derived from the same distribution. The research also imposes a constraint on these data chunks, ensuring that each class  is represented by an equal number of instances in each chunk, thereby examining the impact of restricted data availability. Experiments conducted on CIFAR-10, CIFAR-100, and Tiny-ImageNet demonstrate that, in the context of chunking, most methods perform comparably to standard SGD. Furthermore, the paper suggests a technique called weight averaging, which involves averaging the weights learned at the conclusion of each data chunk.

**Strengths:**

- The paper introduces a novel continual learning (CL) scenario termed "chunking," revealing that in this setting, most techniques exhibit behavior akin to standard SGD. This finding is likely to intrigue the CL community.
- The paper suggested method of weight averaging, inspired by the linear case, is both straightforward and insightful, aiding in comprehending its practical application.
- In terms of clarity, the paper is well-structured, and easy to read.
- Regarding reproducibility, the supplementary materials of the paper include code developed on top of the DER++ framework, enhancing the ease with which future research can replicate these findings.

**Weaknesses:**

- **Unclear motivation and assumptions:** The central motivation surrounding the exploration of limited data availability for CL might benefit from a bit more clarification. Although the paper aims to explore the impact of limited data on CL, the assumptions introduced for 'chunking' in the continual learning framework seem to diverge significantly from practical real-world scenarios. For instance:
   - Assuming data in each chunk are identically distributed negates the occurrence of distribution shifts. In scenarios lacking such shifts and with sufficiently large chunk sizes, performance degradation may be minimal. This could make comparing different methodologies highly sensitive to the chosen chunk size, complicating reproducibility for subsequent research.
  - The constraint that each chunk must contain an approximately equal number of instances for each class is often unrealistic. In real-world scenarios, new categories often emerge over time, and it’s impractical to anticipate equal representation of all classes. Although class imbalance is evaluated in Appendix B, the experimental settings for the imbalance settings is not well articulated.
- **Lack of Distinction from Task-Free Continual Learning:** The paper compares its approach with online continual learning but fails to adequately differentiate it from task-free continual learning approaches [1-4], which do not assume explicit task boundaries. In many situations, identifying shifts in tasks is more critical than defining boundaries for model updating.
- **Weight Averaging**: The method of averaging weights upon the completion of each chunk or task closely resembles existing strategies used in both offline training and online federated learning, raising questions about its innovative aspect. Additionally, this technique presupposes the availability of parameters from all chunks, a condition that might not always be feasible in practical scenarios. Despite this, the paper does not include a comparison with architectural-based continual learning methods (references [5-7]), which is essential to position the performance of this method to methods that make similar assumptions.
- **Comparison with recent methods.** Referencing DER++ as the state-of-the-art method does not align with recent advancements. Comparisons with newer models like GMED (Jin et al., 2021), and CLS-ER (Arani et al., 2022) would be more relevant, considering their similar experimental contexts. Additionally, the paper mentions about the impossibility of resampling data or chunks, but lacks a discussion and incorporation of data-free CL methods [8-10] in the existing literature.
- **Additional minor recommendations.**
    - Table 1 could be improved by including data on smaller memory sizes and adding metrics on forgetting. Additionally, it is essential to have the comparison between different settings in Table 1 for multiple CL methods.
    - Figures should have larger font sizes for enhanced readability.
    - In Section 4.2, the symbols $m,V$ require clear definitions for better comprehension.

**References.**
[1] Aljundi et al., Task-Free Continual Learning. CVPR 2019.
[2] Aljundi et al., Gradient Based Sample Selection for Online Continual Learning. NeurIPS 2019.
[3] Wang et al., Improving Task-free Continual Learning by Distributionally Robust Memory Evolution. ICML 2022.
[4] Ye et al., Task-Free Continual Learning via Online Discrepancy Distance Learning. NeurIPS 2022.
[5] Rusu et al., Progressive Neural Networks.
[6] Yoon et al., Lifelong Learning with Dynamically Expandable Networks. ICLR 2018.
[7] Li et al., Learn to grow: A continual structure learning framework for overcoming catastrophic forgetting. ICML 2019.
[8] Li et al., Learning without Forgetting. ECCV 2016.
[9] Zenke et al., Continual learning through synaptic intelligence. ICML 2017.
[10] Madaan et al., Heterogeneous Continual Learning. CVPR 2022.

**Questions:**

Kindly refer to the concerns raised in the above section.

---

> ### Author Response · Authors · 2023-11-20
> **Author Response**
>
> Dear reviewer, thank you for your comments and questions, we provide answers to them below.
>
> ****1. Why we look at the chunking setting and not a more realistic setting****
>
> We agree with you that the chunking setting is not a suitable model for many real-world scenarios and state as much already in the paper ([Page-2, Section-2, Para-3]). However, this misses the point of the paper which is that any more realistic CL setting with task/distribution shift can be decomposed into two sub-problems, one of which is the chunking problem. We show that the chunking problem is a significant part of the most common CL settings and has been underexplored as no current CL methods performs better than SGD on it. Therefore, a way to improve on any real-world CL problem would be to improve performance on the chunking problem, which is the key point of our work and is currently underexplored. We hoped that this point was clear in the paper but given your comment we see that we need to be clearer on this point and will edit the paper accordingly.
>
> Also, in your review you mention the use of class-balanced chunks, this is to ensure that our experimental findings are due to the chunking problem and not class imbalance. However, we agree with you that we could describe better the random-sampling experimental setting looked at in appendix B and have edited the paper accordingly.
>
> ****2. Difference between task-free CL and the chunking setting****
>
> In task-free CL, as you mention, there is distribution shift but unlike other settings the distribution changes slowly through time. This is unlike the chunking setting where we look only at the problems arising from seeing the data in chunks and so by design contains no distribution shift. Therefore, the settings are quite different. However, like other CL settings, as we see the data in chunks, chunking is a subproblem of task-free CL and so improving performance in the chunking setting is necessary for task-free CL performance to approach that of offline performance.
>
> ****3. Reason we look at per-chunk weight averaging****
>
> In your review you point out that weight averaging has been used in offline learning and federated learning before and hence question its general novelty. We agree with this point and already state so in the paper ([Page-3, Section-2, Para-4]). However, we hope that you agree with us that it has not been looked at as a method to improve performance in the chunking setting, which is the reason we look at it. That is, we look a per-chunk weight averaging to show that you can improve performance and reduce forgetting in the chunking setting and that this performance improvement transfers to full CL settings where there is also task/distribution shift. Therefore, we hope you now can see that in the context of chunking by looking at per-chunk weight averaging we have provided a useful starting point for improving performance in the chunking setting.
>
> ****4. Incorporation of data-free methods and other recent methods****
>
> We do look at a data free method in our experiments on the chunking setting, EWC and have run experiments with SI where it had the same behaviour as EWC in the chunking setting. Additionally, we thank you for pointing out the two CL methods GMED and CLS-ER, we agree with you that adding CLS-ER to our chunking experiments would be beneficial and will do so but this might take some time. However, in the case of GMED it is only designed for the online CL so can not be used for any chunk size greater than that of a mini-batch and so can not be readily added to our chunking experiments. Also, we would like to say that both papers focus on the challenge of dealing with task/distribution shift and do not mention or give thought to the other sub-problem of CL, the chunking problem. This is the main point we are trying to put across in our paper, that the chunking problem has been overlooked as a sub-problem in CL and hence these works do not challenge the main point of this paper, instead they strengthen it.
>
> ****5. Minor recommendations****
>
> Thank you for these points, we will edit the paper accordingly.
>
> ****
>
> We hope that given our answers to your questions that we have shown the reason why the chunking setting and analysis of it is of importance to the CL community and that you will consider our answers when you reconsider your review.

---

### Official Review · Reviewer_4UBT · 2023-10-29

**Soundness:** 2 fair
**Presentation:** 2 fair
**Contribution:** 2 fair
**Rating:** 3
**Confidence:** 4

**Summary:**

The paper considers a setting of continual learning called "chunking".

The paper argues that chunking is an important setting and overlooked by prior work.

The paper presents experimental analysis and a weight-averaging method for the chunking setting.

**Strengths:**

Omitted

**Weaknesses:**

The paper has several weaknesses:
- First, it does not seem to be a solid work. What the paper does is essentially as follows:
   - Take the CL library *Mommoth*.
   - Run all methods there in the specific setting of chunking. In my eyes this would require only a minor modification to load the data in a different way.
   - Propose a method that can be implemented in a few lines.

- The chunking setting is questionable:
   - First, in Figure 1, I didn't see a difference between the so-called chunking setting and domain-incremental learning, where the change of task is only indicated by the change of data. In the language of chunking, each chunk is a task in domain-incremental learning.
   - The paper argues that there is no task shift in chunking (while domain-incremental learning assumes a task shift like rotation or permutation of the original data). However, I didn't find this argument convincing, as task shift is hard to measure. For example, I am not sure which of the following has a larger task shift: 1. from digit 1 to digit 4 (chunking); 2. from digit 1 to a rotated digit 1 (domain-incremental learning).
   - In summary, the notation of chunking appears to be a word game and I couldn't see the significance on its own or difference from prior works.

Based on the above understanding, I don't find the experiments of interest. I consider chunking to be an instance of domain-incremental learning, and the experimental results are natural consequences of domain shift and the specific training setups. In particular:
- In Figure 2, I am surprised to see that the chunk size is reduced from $>10^4$ to $10^2$, while the paper trains it for 50 epochs all the time. In other words, the model is given $10^2$ samples for each task, and trained for $50$ epochs. I think this would easily lead to overfitting and It would only be surprising if it does not forget the previous knowledge. Moreover, for the chunk size of $10^2$ I am not sure how large the test set is. I would guess the corresponding test set has much fewer samples, and therefore, it is easily biased rather than being representative on the performance over the entire distribution.
- What distinguishes SGD and other (memory-based) CL methods is the memory buffer size. However, the paper does not seem to evaluate the effect of memory sizes. I would argue that for larger memory sizes, the performance difference between SGD and CL methods would appear. Therefore, in my eyes, what the paper shows is that, for small chunk sizes (or "equivalently" for increasingly many tasks and fewer samples per task), the current CL methods are less powerful, and would require more memory to resist forgetting. And if my reasoning is correct, this paper would be very incremental and would contribute in a very limited way.

Finally, The performance on Cifar100 and Tiny ImageNet in Table 2 does not look good. For example, the table presents an accuracy of 5.48 for DER++, and the weight averaging improves this to 8.53. The accuracy is still smaller than 10%. Can we claim weight averaging gives a victory? From this table, I can only reason that none of the methods works in the proposed setting, including weight averaging.

**Questions:**

I have no questions.

---

> ### Author Response · Authors · 2023-11-20
> **Author Response (Part 1)**
>
> Dear reviewer, thank you for your comments which we answer below.
>
> ****1. Difference between Domain incremental learning and the chunking setting****
>
> From your review it seems your main criticism is that you think the chunking setting is the same as domain incremental learning (DIL). However, the chunking setting and DIL are quite different settings. This is because by definition DIL has a task/distribution shift element, as p(x) is different for each task; while the point of the chunking setting is to only look at the chunking sub-problem and not the task-shift element of CL and so by construction it has no task/distribution shift element. Also, experiments from previous work show that in DIL methods like DER++ and ER perform much better than SGD (e.g., Buzzega et. al 2020). This is very different from the chunking setting where one of our main findings is that no current CL method we tested performed better than SGD. Therefore, we hope that you can see that the chunking setting is clearly different from DIL. Additionally, we would like to emphasise here why the chunking setting is important to CL, including for DIL. The chunking setting is a sub-problem of CL, including DIL, which is not currently solved or looked at and it is necessary to improve performance on it for CL methods to approach the performance of offline learning.
>
> ****2. Setting the number of epochs for different chunk sizes****
>
> We validated what number of epochs to use for each chunk size and choose the number of epochs which gave the best performance (50 for CIFAR-10 and CIFAR-100 and 100 for Tiny ImageNet). This is described in Appendix C, however we notice that we did not mention this in the main paper and so thank you for pointing it out as we have now added mention of it in the main paper.
>
> Additionally, in your comment you guess that the size of the test set scales with the chunk size. This is incorrect as the size of the test set is fixed for any chunk size and is the standard test set used for the datasets used in the paper. We hoped to of explained this in Section 3 but given your comment we have edited the paper to make it clearer.
>
> ****3. Memory buffer sizes for replay methods****
>
> As part of your review you mention the effect that the size of the memory buffer for replay methods has in the chunking setting. In our experiments, we look at standard memory sizes used by the CL community (e.g., Caccia et al., 2021, and Buzzega et. al 2020). By comparing the chunking results in Table 1 to Figures 2 and 3 you can see that increasing the memory size does not improve performance that much when you split CIFAR-100 or Tiny ImageNet into 10 chunks. Therefore, for standard memory sizes current CL methods performances are comparable to SGD in the chunking setting. However, we agree with you that for much large memory sizes than those commonly used in the CL community that there will be a performance gap between SGD and replay based methods (at some point this must occur as given space to store all the data you can easily reach offline training performance). Additionally, we agree further that this is not a interesting insight, as storing examples effectively inflates the chunk size which is why we do not mention it in the paper. However, what is interesting is that unlike in the full CL setting where replay methods have a range wide range of performances in the chunking setting they all have roughly the same performance. This shows that the improvement over the simplest replay method ER (and for SGD with standard memory sizes) is due to methods improving at the task shift part of CL and not the chunking part. Therefore, we have shown that the current improvement we see in CL, at least for the methods we tested, is due to improving in dealing with task shift, while the chunking problem has been untouched. Hence, an underlooked and significant way (given Table 1) to improve CL performance is to see how we can improve performance on the chunking problem, which is a key take away for our paper and believe is of interest to the CL community.

---

> ### Author Response · Authors · 2023-11-20
> **Author Response (Part 2)**
>
> ****4. Performance of methods in class-incremental learning****
>
> You mention in your review that for online class-incremental learning (CIL) for TinyImageNet the performance of DER++ with and without per-chunk-weight averaging is not very high, being under 10\%. We agree with you; online CIL is a challenging setting which has not been solved to a good degree yet. However, this misses the point of Table 2, which is to empirically show that a method which is designed to improve performance in the chunking setting can also improve performance in full CL settings, where there is task-shift.
>
> ****
>
> We hope that given our rebuttal you can see that our work is not trivial or incremental but is instead bringing light to the overlooked chunking sub-problem of CL. Where we have shown that current CL methods do not solve the chunking problem and that it is necessary to solve for CL methods to achieve the aim of having performance approaching that of offline learning.
>
> ****
>
> Pietro Buzzega, Matteo Boschini, Angelo Porrello, Davide Abati, and Simone Calderara. Dark Experience for General Continual Learning: a Strong, Simple Baseline. Proceedings of the 33rd Conference on the Advances in Neural Information Processing Systems, 33:15920–15930, 2020
>
> Lucas Caccia, Rahaf Aljundi, Nader Asadi, Tinne Tuytelaars, Joelle Pineau, and Eugene Belilovsky. New Insights on Reducing Abrupt Representation Change in Online Continual Learning. In Proceedings of the 10th International conference on Learning Representations, 2021.

---

### Official Review · Reviewer_w2jf · 2023-10-30

**Soundness:** 2 fair
**Presentation:** 3 good
**Contribution:** 3 good
**Rating:** 3
**Confidence:** 4

**Summary:**

The paper presents an analysis of the 'chunking' sub-problem in continuous learning (CL), a problem that has received limited attention in the existing literature. The authors argue that chunking is responsible for a significant part of the performance drop during CL, and forgetting is responsible for the performance drop in chunking. They also demonstrate that existing CL algorithms are ineffective for the forgetting in chunking. Leveraging insights from linear models, the authors propose a per-chunk weight averaging method, which significantly improves performance in a chunking scenario and remains effective in the full CL setting.

**Strengths:**

The paper tackles a relatively overlooked problem in the continue learning literature - the 'chunking' problem. The chunking problem is a separate problem and different from the distribution shift problem typically assumed in most continue learning literature. Furthermore, chunking exists in all continue learning and online learning settings, therefore deficiencies caused by chunking could affect all such scenarios. This means that a better understanding of the chunking problem could have wide implications for continual learning and relevant problems.

This work identifies a significant deficiency in learning caused by chunking the data alone. The authors challenge conventional wisdom that forgetting is mainly caused by distribution shifts. The analysis presented in the paper provides valuable insights into the chunking problem, and potentially draws attention to developing different kinds of effective algorithms beyond the usual distribution shift paradigm.

The proposed per-chunk weight averaging is simple and effective, well-supported with empirical evidence.

**Weaknesses:**

* One of the paper's main conclusions "forgetting is the main reason for performance drop in chunking setting" is not logically supported by evidence. The authors observed forgetting on the training set in Figure 5 and a performance drop on the test in Figure 2, and claim that forgetting (on the training set) leads to a performance drop (on the test set), which does not appear to have a causal relationship.

  It is shown in Figure 5 that there is significant forgetting of the overfitting features on previous training chunks, but it is not clear whether the learned generalizable features are forgotten (previous chunk accuracy never dips below test accuracy). Learning overfitting features is not causal for generalization performance, as one can often train good-performing models with little overfitting by adjusting hyperparameters and/or using regularization. Therefore, forgetting of overfitting features is also not causal for losing generalization performance.

  There is a simple explanation of the observed forgetting on past chunks: overfitting features are, by definition, unique to a specific combination of training examples, so overfitting features learned in one chunk are not experienced again in subsequent chunks, leading to their forgetting. Generalizing features, however, are repeatedly experienced in multiple chunks and should manifest much lower forgetting (at least in theory). Ultimately, we are concerned about the forgetting of generalizing features rather than overfitting features, as the former determines the final generalization performance.

* The proposed weight averaging method already exists in the CL literature, as the Incremental Moment Matching method proposed in [1]. The Mean-IMM method in [1] is essentially weight averaging, and significant performance improvement is reported when applied to CL tasks. In light of [1], it is probably safer to say "most popular CL algorithms ... only performing as well as SGD" rather than "current CL algorithms" in the abstract of the current paper.

  [1] Lee, Sang-Woo, et al. "Overcoming catastrophic forgetting by incremental moment matching." *Advances in neural information processing systems* 30 (2017).

**Questions:**

* How would the Adam optimizer perform compared to plain SGD? If the moment estimations of Adam optimizer persist across chunks, maybe it could have a similar smoothing effect similar to weight averaging.
* Epoch numbers and learning rates are set to the same for different chunk sizes: I personally don't think that this is a very fair setting as the author believes. When training many epochs on a small chunk, the model will see the same examples more frequently and with shorter intervals, which could lead to more severe overfitting and negatively impact generalization. I would suggest finding optimal hyperparameters separately for different chunk sizes (maybe divide all chunk sizes into several ranges and find one set of hyperparameters for each range).
* In my opinion (offering to the authors for pure discussion), the chunking problem may not be so different from a distribution shift, at least for small chunk sizes. With smaller chunks, the difference in the chunks' examples statistics will increase, even if examples are all sampled from the same distribution. For example, when the chunk size is small, some chunks will inevitably happen to have more images with indoor backgrounds, and some happen to have more outdoor backgrounds. This could be impossible to tell from true distribution change, and the model could still struggle with the changing examples statistics.

---

> ### Author Response · Authors · 2023-11-20
> **Author Response**
>
> Dear reviewer, we thank you for your questions and comments. We provide answers to them below.
>
> ****1. Relation between forgetting on previous chunks and final performance****
>
> Yes, we agree with you we could be clearer on how the forgetting of a chunks training data impacts performance. However, there is a relationship as it is expected that training performance should be roughly greater than or equal to test performance, as the learner has trained to fit the training data well (this is precisely true, in expectation, for empirical risk minimisation methods and is true for our experiments). Therefore, given that the performance on the training data of a chunk quickly drops to a low level after learning on it for standard CL methods (i.e., they forget), in the near future the test performance can be no better than this training performance. Hence, the drop in training performance of a chunk after learning on it, that is the amount of forgetting, affects the current possible best case test performance. We also show that per-chunk weight averaging gets much better performance in the chunking setting than the CL methods we tested and it also forgets a lot less, giving experimental evidence to the claim also. We hope now that we have satisfactorily answered your question on this matter and thank you for it as we think it will strengthen our submission.
>
> ****2. Incremental Moment Matching****
>
> Thank you for pointing out the work on IMM, we agree with you that IMM is very similar to per-chunk weight averaging (though not the same as IMM affects training behaviour while per-chunk weight averaging is designed not to affect training at all). However, the general novelty of per-chunk averaging is not why we include it in the paper, instead we include it to show that you can using simple methods improve performance in the chunking setting and that this performance transfers to the full CL setting, where task-shift occurs. This insight has not been presented before and so in this context we believe that looking at per-chunk weight averaging is of interest to the CL community.
>
> ****3. Using Adam optimiser instead of SGD****
>
> In our exploration of the best learning setup we did look at using momentum, weight decay and other optimisers. However, we found the best performance across all methods was to use the setup that we use in the main paper, which is also the same as previous work (Buzzega et. al 2020). Additionally, we note that in our experience of CL using Adam usually performs worse than SGD.
>
> ****4. Setting of number of epochs and learning rate for different chunk sizes****
>
> We validated what number of epochs to use for each chunk size and choose the number of epochs which gave the best performance (50 for CIFAR-10 and CIFAR-100 and 100 for Tiny ImageNet), as described in Appendix C. Additionally, we validated our choice of learning rate and again saw that setting it two the same value for each chunk size gave the best performance, as described in Appendix A. However, we notice that we did not mention this in the main paper and so thank you for pointing it out as we have now added mention of it in the main paper.
>
> ****5. Distribution shift vs empirical difference in samples drawn from a i.i.d. process****
>
> We agree with you that small changes in distribution can be imperceptible from the differences between samples drawn from the same distribution (e.g., the difficulty of two sample testing). Additionally, we agree with you that the difficulty of the chunking setting comes from the empirical differences between chunks. However, this is not the same kind of shift that CL methods currently aim to tackle as shown by the fact the ones we test do not perform any better than SGD in the chunking setting. Therefore, there is a gap in the literature for researchers to solve this kind of shift and we show that by doing so it is likely to improve performance in the full CL setting, as chunking is a sub-problem of any CL setting. Lastly, I hope the reviewer agrees with us that the difference between i.i.d. samples and non-stationary processes can be quite distinct in theoretical results.
>
> ****
>
> We hope that given our answers to your questions that you can reconsider your review. Also, we note that it seems to us by reading your review that you do not have a problem with the main point of the paper; that the chunking problem is an underexplored significant part of CL and is necessary to solve before we can solve CL in general.
>
> ****
>
> Pietro Buzzega, Matteo Boschini, Angelo Porrello, Davide Abati, and Simone Calderara. Dark Experience for General Continual Learning: a Strong, Simple Baseline. Proceedings of the 33rd Conference on the Advances in Neural Information Processing Systems, 33:15920–15930, 2020

---

### Official Review · Reviewer_T7s5 · 2023-11-03

**Soundness:** 2 fair
**Presentation:** 3 good
**Contribution:** 2 fair
**Rating:** 5
**Confidence:** 4

**Summary:**

- This work investigates the process of grouping the data in chunks i.e., chunking, in online continual learning (CL) settings where data is encountered as a non-stationary stream and hence past seen data cannot be revisited anymore.
- In doing so, the authors demonstrate that the chunking is responsible for almost half of the performance drop due to catastrophic forgetting which has mainly been associated with the drift in data distribution in recent CL literature.
- Extensive experimental analysis of chunking in online CL settings shows that: (a) forgetting reduces as chunk size increases, (b) forgetting is the main reason for the performance drop, and (c) per-chunk weight averaging improves model performance thus leading to less forgetting.

**Strengths:**

- Originality
	- The authors have presented the analysis of online CL settings from the novel perspective of data grouping i.e., chunking. Additionally, a simple method of "per-chunk weight averaging" has been proposed that minimizes forgetting of the previously seen chunks by averaging the weights after training is finished for each chunk.
- Quality
	- The motivation is well-founded.
- Clarity
	- Paper is clearly presented.
	- Mathematical formulation is detailed and explanatory.
- Significance
	- The proposed "per-chunk weight averaging" (WA) method significantly outperforms the baseline methods, which demonstrates the efficacy of this method in improving performance by retaining past knowledge and minimizing forgetting.

**Weaknesses:**

- Originality
	- Contributions are incremental and novelty is limited.
- Quality
	- [Page-1, Section-1, Para-3] When a model sequentially learns over a sequence of datasets (in this case chunks) without having the measures of retaining the past knowledge it tends to forget the past learning as evident in the CL literature in both cases of homogeneous and heterogeneous data (chunk) distributions. Therefore, it is expected that the model performance will drop in such cases, hence the second claim is a valid expectation in such a setting and does not constitute a significant outcome of the analysis.
	- In practical scenarios where task boundaries are not pre-known or specified, per-chunk weight averaging could easily worsen the model performance as averaging is done without consideration of the current chunk's domain/distribution as compared to the past chunks.
	- Per-chunk weight averaging can be seen as the simplest form of knowledge aggregation technique in a continual learning setting. Hence, it reduces forgetting in the ideal (class-balanced) scenario which is the usual expectation from such techniques and does not constitute high significance as more sophisticated methods have already been developed in CL literature like weight regularization, data replay and incorporating additional parameters.
	- The chunking setting described in the papers is ideal and data for such settings is also generated under simplified and impractical assumptions which will not scale to the online settings with changing data distributions in real-world scenarios.
	- I am keen to hear the response of the authors on this and hope that they can change my point of view.
- Clarity
	- It is difficult to keep track of the different data preparation techniques for "offline SGD", "standard CL" and "chunking" methods. It would be better to have clear algorithms and/or pictorial illustrations for the same.
	- [Page-2, Section-2, Para-2] Please elaborate on the classification problems being referred to here.
	- [Figure-2] The Font size is too small, please increase it and also add an explanation of the models shown in the legend (Move them from Appendix A to the Figure caption).
	- Inconsistent use of terminologies. Please clarify the following terminologies (maybe in a tabular format) so that the reader can refer to them whenever required:
		- "offline learning"
		- "plain SGD learning"
		- "full CL setting"
		- "standard CL"
		- "online CL"
- Significance
	- [Table-2] As "per-chunk weight averaging" strategy involves updating weight parameters. It would make sense to compare it with the existing "weight regularization" based CL strategies like the below methods:
		- [EWC] -> Kirkpatrick, James, Razvan Pascanu, Neil Rabinowitz, Joel Veness, Guillaume Desjardins, Andrei A. Rusu, Kieran Milan et al. "Overcoming catastrophic forgetting in neural networks." Proceedings of the national academy of sciences 114, no. 13 (2017): 3521-3526.
		- [SI] -> Zenke, Friedemann, Ben Poole, and Surya Ganguli. "Continual learning through synaptic intelligence." In International conference on machine learning, pp. 3987-3995. PMLR, 2017.
- Typographical errors:
	- [Page-1, Section-1, Para-1] "thwart" -> "thwarted", "focuses CL" -> "focuses of CL"
	- [Page-2, Section-2, Para-2] "called called" -> "called"

**Questions:**

- [Page-2, Section-2, Para-2] How does one decide how many chunks (mini-batches) are there in a particular task if there is no pre-defined task boundary in online CL setting?

---

> ### Author Response · Authors · 2023-11-20
> **Author Response**
>
> Dear reviewer, thank you for your review, we provide answers to your comments and questions below.
>
> ****1. Performance drop due to chunking****
>
> We agree that the observation that learning over a stream of chunks results in a performance drop from offline learning is not surprising. However, we do think the finding that this performance drop is large enough to be a significant part of the performance drop of CL from offline learning is an important finding. This is because it suggests that a necessary step towards solving CL in general is improving the performance in the chunking setting. Additionally, given our finding that current CL methods do not improve performance over SGD in the chunking setting, we have shown that the chunking problem has been overlooked and is necessary to solve CL in general.
>
> ****2. Reasons for looking at per-chunk weight averaging****
>
> In your review you ask why we look at per-chunk weight averaging in the chunking setting and not a more complex knowledge aggregation method already proposed in the CL literature. This is because one of our key findings is that current knowledge aggregation CL methods (EWC, DER++, ...) do not perform any better than plain SGD learning in the chunking setting. Therefore, to improve performance in the chunking setting we had to start from scratch and from an analysis of the linear case we saw that per-chunk weight averaging could improve performance, which we empirically demonstrate.
>
> Also, you comment that in CL settings where task-boundaries are not known per-chunk weight averaging could hurt performance. This point might certainly be true. However, the reason we look at per-chunk weight averaging is to show that we can improve performance in the chunking setting (where there are no task boundaries) and that this improved performance in the chunking setting can transfer to improved performance in the most common CL settings. Therefore, we see your point as valid but not in the scope of this work as in this work we focus on exploring the benefits of looking at the chunking setting and not how to adapt methods to CL settings without task boundaries (but with distribution shift).
>
> ****3. Reason why we look at the chunking setting****
>
> In your review you state that the chunking setting is not a suitable model for many real-world scenarios. We agree with this statement and state as much already in the paper ([Page-2, Section-2, Para-3]). However, this misses the point of the paper which is that any more realistic CL setting with task/distribution shift (including ones without task boundaries) can be decomposed into two sub-problems, one of which is the chunking problem. We show that the chunking problem is a significant part of the most common CL settings and has been underexplored as no current CL methods performs better than SGD on it. Therefore, a way to improve on any real-world CL problem would be to improve performance on the chunking problem, which is a key point of our work and is currently underexplored.
>
> ****4. Use of terminology (e.g. offline, standard CL, online CL, plain SGD learning and the full CL setting)****
>
> Thank you for pointing this out. While in the paper we have whenever we introduce a term from the CL literature introduced what it means, from your comment we see that this could be clearer and so have edited the paper accordingly.
>
> ****5. Typographical errors****
>
> Thank you for pointing out these typos, we have fixed them in the paper.
>
> ****6. [Page-2, Section-2, Para-1] How does one decide how many chunks (mini-batches) are there in a particular task if there is no pre-defined task boundary in online CL setting?****
>
> We are confused by what you mean in this question as we do not know what you mean by "there is no pre-defined task boundary in online CL setting". In online CL you see each task in turn and for each task you see a sequence of mini-batches where you see each mini-batch once and cannot revisit previous mini-batches, as shown in Figure 1. Therefore, there are task boundaries in online CL and the number of mini-batches you see per-task is dependent on the dataset and how you split the dataset into tasks.
>
> ****
> We hope that when reconsidering your review you take into account our answers to your questions above, especially on why looking at the chunking setting is useful.

---

### Meta-Review · Area_Chair_beZu · 2023-12-09

**Metareview:**

The paper focuses on the issue of forgetting in continual learning due to mini-batch updates without replay buffers, referred to as chunking. The paper shows that the issue occurs even in single-task-based learning. The work proposed performing a per-chunk weight averaging to improve performance.

The thorough reviews and post-rebuttal comments give a common impression that the paper is not ready for publication.

Strengths:

- Focus on the forgetting issue in a single-task setup as opposed to the common setup of distribution shift under multiple tasks.

Weaknesses:

- While the chunking setup attempts to highlight the forgetting issue in a single-task setup, the setup contains unrealistic assumptions without a clear connection to a realistic setup. The confusion regarding the role of this setup in elucidating an issue as opposed to portraying a realistic issue was apparent in the reviewers’ comments.

- The work also lacks in terms of the solution proposed. The per-chunking weight averaging can be seen as an illustration of how the issue can be addressed with a simple method, but the reviewers mentioned its similarity with existing methods, and the authors acknowledged that the solution is meant to be a simple illustration of a simple method as opposed to a novel approach.

- While conceptual or theoretical insights could still be interesting, the shortcomings of the provided results, such as an accuracy of 8.53 on Tiny ImageNet being considered as an improvement, leave much to be desired.

**Justification For Why Not Higher Score:**

In the list of weaknesses above, I clarified how the work has significant shortcomings regarding experiments, the clarity of the new setup, and the originality of the proposed solution.

**Justification For Why Not Lower Score:**

N/A

---

### Decision · Program_Chairs · 2024-01-16

Reject